# Fundamentals of Lossless, Reciprocal Bianisotropic Metasurface Design

Luke Szymanski , Brian O. Raeker , Chun-Wen Lin and Anthony Grbic *,†

Department of Electrical Engineering and Computer Science, University of Michigan, Ann Arbor, MI 48109, USA; ljszym@umich.edu (L.S.); braeker@umich.edu (B.O.R.); chunwen@umich.edu (C.-W.L.)
* Correspondence: agrbic@umich.edu
† Current address: 1301 Beal Avenue, EECS Building, Room 3238, Ann Arbor, MI 48109, USA.

**Abstract:** Lossless, reciprocal bianisotropic metasurfaces have the ability to control the amplitude, phase, and polarization of electromagnetic wavefronts. However, producing the responses that are necessary for achieving this control with physically realizable surfaces is a challenging task. Here, several design approaches for bianisotropic metasurfaces are reviewed that produce physically realizable metasurfaces using cascaded impedance sheets. In practice, three or four impedance sheets are often used to realize bianisotropic responses, which can result in narrowband designs that require the unit cells to be optimized in order to improve the performance of the metasurface. The notion of a metasurface quality factor is introduced for three-sheet metasurfaces to address these issues in a systematic manner. It is shown that the quality factor can be used to predict the bandwidth of a homogeneous metasurface, and it can also be used to locate problematic unit cells when designing inhomogeneous metasurfaces. Several design examples are provided to demonstrate the utility of the quality factor, including an impedance matching layer with maximal bandwidth and a gradient metasurface for plane wave refraction. In addition to these examples, several metasurfaces for polarization control are also reported, including an isotropic polarization rotator and an asymmetric circular polarizer.

**Keywords:** metasurfaces; bianisotropy; metasurface bandwidth

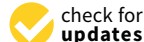



## 1. Introduction

Metasurfaces are the two-dimensional analogue of metamaterials, which interact with electromagnetic fields at a surface rather than throughout a volume [1]. They are often realized using electrically thin layers that consist of 2D arrays of subwavelength-spaced meta-atoms that can be homogenized. This allows for metasurfaces to be modeled using surface boundary conditions, called the generalized sheet transition conditions (GSTCs), which determine the interaction with an incident field through quasi-static surface polarizabilities [2–4]. If the incident fields are time-varying, then the surface polarizabilities produce equivalent electric and magnetic polarization currents. These equivalent currents can be related to surface admittances and impedances [5–11]. The mechanisms by which electric and magnetic surface currents are induced on a metasurface determine its classification as an electric, magnetic, electric and magnetic, or bianisotropic metasurface. Purely electric metasurfaces only contain electric polarizabilites, which interact with only the electric field to produce electric currents. Purely magnetic metasurfaces only contain magnetic polarizabilities, which only interact with the magnetic field to produce magnetic currents. Electric and magnetic surfaces contain both electric and magnetic polarizabilities in a single sheet. Finally, a metasurface that is bianisotropic can also contain electric and magnetic polarizabilities, as well as electro-magnetic and magneto-electric polarizabilities, i.e., magnetic polarization due to an electric field and electric polarization due to an magnetic field.

Bianisotropic metasurfaces provide the metasurface designer with the most degrees of freedom, which makes them useful for the extreme manipulation of electromagnetic fields. Bianisotropic metasurfaces include a wide range of both reciprocal and non-reciprocal responses. However, the focus here will be on design methods for reciprocal bianisotropic metasurfaces. Reciprocal bianisotropic metasurfaces can be split into two main classes: chiral and omega. Chiral metasurfaces contain meta-atoms that have broken mirror symmetry. This results in electric fields inducing magnetic currents along the impinging electric field and magnetic fields inducing electric currents along the impinging magnetic field. These chiral responses alter the polarization state of the incident wave. On the other hand, omega metasurfaces contain meta-atoms with broken directional symmetry. This results in electric fields inducing magnetic currents that are orthogonal to the impinging electric field, and magnetic fields inducing electric currents orthogonal to the impinging magnetic field. This leads to an asymmetric scattering response from omega metasurfaces, which maintains the polarization state.

The applications for chiral and omega bianisotropic metasurfaces fall into two main categories: those that guide and radiate electromagnetic waves and those that control reflection and transmission from a surface. Guided-wave bianisotropic metasurfaces shape fields along the surface through guided or leaky waves, and they can be used to produce desired radiation patterns [12,13]. Whereas, metasurfaces that control reflection and transmission interact with incident wavefronts to manipulate the amplitude, phase, and polarization of the scattered fields. There are many design synthesis methods and realizations of planar and cylindrical bianisotropic metasurfaces that control reflection and transmission at frequencies that range from microwave to optical using both composite (metal/dielectric) and all-dielectric metasurfaces [5–7,11,14–30]. This is by no means a complete representation of all the work in bianisotropic metasurfaces. For a more complete review of the literature, see [31].

In this paper, the design and synthesis methods that are presented in [5,23] are reviewed, and several design examples are provided. Additionally, a definition for the quality factor of a three-sheet metasurface is provided, which can be used to estimate the bandwidth of a homogeneous metasurface. This is demonstrated through the design of an impedance matching metasurface with maximal bandwidth. In addition to improving bandwidth, the quality factor can also aid designers in improving the performance of inhomogeneous metasurfaces. This is demonstrated by using the quality factor to guide the selection of appropriate unit cells in the design of a gradient metasurface for plane wave refraction.

## 2. Scattering from Bi-Isotropic Metasurfaces

In this section, we describe the scattering performance of an omega-type bi-isotropic metasurface that is illuminated by a normally incident plane wave. The scattering analysis of bi-isotropic metasurfaces provided in this section follows that introduced in [5]. We consider a metasurface at a planar boundary between two regions of space, as shown in Figure 1, where the intrinsic wave impedance of region 1 is $\eta_1 = \sqrt{\mu_1/\epsilon_1}$ and of region 2 is $\eta_2 = \sqrt{\mu_2/\epsilon_2}$. The metasurface is at the $z = 0$ plane separating the two regions, and it is illuminated by normally incident plane waves.

The interaction between the metasurface and an illuminating plane wave can be described via scattering parameters (S-parameters), which comprise the ratio between the scattered plane wave electric field and the incident plane wave electric field. The ratio of scattered electric field in region $n$ to the incident electric field in region $m$ for different polarizations is given as a $2 \times 2$ matrix.

$$S_{nm} = \begin{pmatrix} S_{nm}^{xx} & S_{nm}^{xy} \\ S_{nm}^{yx} & S_{nm}^{yy} \end{pmatrix} \tag{1}$$

When viewed from region 1, $S_{11}$ is the reflection coefficient and $S_{21}$ is the transmission coefficient. Similarly, when viewed from region 2, the reflection coefficient is $S_{22}$ and the transmission coefficient is $S_{12}$.

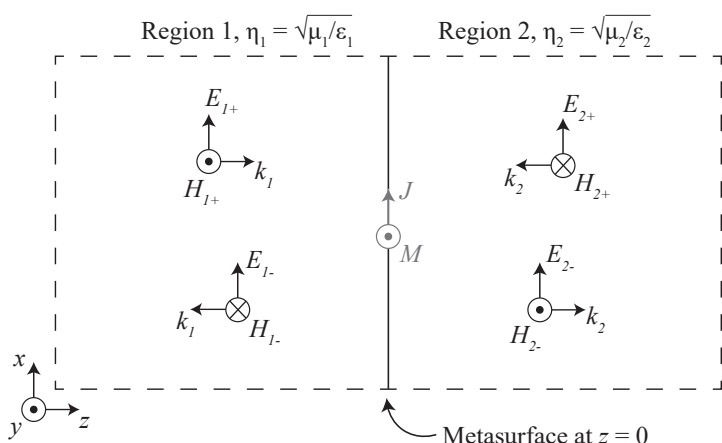

**Figure 1.** The geometry of a metasurface between two regions with different material properties. The equivalent surface current densities $J$ (electric) and $M$ (magnetic) describe the interaction of the metasurface with the tangential fields. Under illumination by a normally incident plane wave, each region can contain two plane waves that are denoted by $+$ for a wave propagating toward the surface or $-$ for a wave propagating away from the surface.

In general, a bianisotropic metasurface can be modeled as a two-dimensional array of polarizable particles [4]. For time-varying illuminating fields, the polarizabilities can be effectively characterized using equivalent surface impedances [5–11]. The equivalent surface currents can then be related to the averaged, tangential electric, and magnetic fields using surface parameters that are represented as $2 \times 2$ tensors: the electric sheet admittance tensor $Y$, the magnetic sheet impedance tensor $Z$, and the magneto-electric coupling tensors $\chi$ and $\gamma$. With these parameters, the electric and magnetic surface currents that are induced on the metasurface can be related to the average tangential fields and compared to the boundary conditions across the metasurface.

$$\begin{pmatrix} J \\ M \end{pmatrix} = \begin{pmatrix} Y & \chi \\ \gamma & Z \end{pmatrix} \begin{pmatrix} E_{avg} \\ H_{avg} \end{pmatrix} = \begin{pmatrix} \hat{z} \times (\bar{H}_2 - \bar{H}_1) \\ -\hat{z} \times (\bar{E}_2 - \bar{E}_1) \end{pmatrix} \tag{2}$$

The variables $Y$, $\chi$, $\gamma$, and $Z$ relate the $x$- and $y$-polarized averaged field components to the $x$- and $y$-polarized current density components that are induced on the metasurface. The various electric field vectors are $E = \begin{bmatrix} E_x & E_y \end{bmatrix}^T$ and the magnetic field vector is $H = \begin{bmatrix} H_x & H_y \end{bmatrix}^T$ (the surface current quantities $J$ and $M$ are similarly defined), where the averaged fields are $E_{avg} = (E_1 + E_2)/2$ and $H_{avg} = (H_1 + H_2)/2$. The electric admittance tensor is defined as

$$Y = \begin{pmatrix} Y_{xx} & Y_{xy} \\ Y_{yx} & Y_{yy} \end{pmatrix} \tag{3}$$

with the other parameters being similarly defined.

For a reciprocal metasurface, $Y = Y^T$, $\gamma = -\chi^T$, and $Z = Z^T$ [32]. Imposing isotropy on the surface parameters results in

$$Y = YI, \qquad \chi = -\chi n, \qquad \gamma = -\gamma n, \qquad Z = ZI \tag{4}$$

where $I$ is the $2 \times 2$ identity matrix and $n = \begin{pmatrix} 0 & -1 \\ 1 & 0 \end{pmatrix}$.

Restricting the metasurface to omega-type bi-isotropy precludes polarization conversion by the metasurface. Therefore, the response for each polarization is identical. This allows us to analyze the metasurface as a two-port network for a single polarization, rather

than as a four-port network when all of the polarizations were considered. The two-port S-parameters relate the electric field of the incident and reflected plane waves as

$$\mathbf{E}_- = \begin{pmatrix} E_{1-} \\ E_{2-} \end{pmatrix} = \begin{pmatrix} S_{11} & S_{12} \\ S_{21} & S_{22} \end{pmatrix} \begin{pmatrix} E_{1+} \\ E_{2+} \end{pmatrix} = \mathbf{S}\mathbf{E}_+ \tag{5}$$

In order to calculate the S-parameters of the metasurface, consider an $x$-polarized plane wave, as shown in Figure 1. Assuming that the surface is isotropic, the boundary conditions of Equation (2) simplify to

$$J_x = \frac{Y}{2}(E_{1x} + E_{2x}) + \frac{\chi}{2}(H_{1y} + H_{2y}) = -H_{2y} + H_{1y} \tag{6}$$

$$M_y = -\frac{\gamma}{2}(E_{1x} + E_{2x}) + \frac{Z}{2}(H_{1y} + H_{2y}) = -E_{2x} + E_{1x} \tag{7}$$

From Equations (6) and (7), we obtain four equations by considering the illumination from region 1 ($E_{2+} = 0$) and region 2 ($E_{1+} = 0$) separately. These four equations relate the S-parameters to the surface parameters of the metasurface. In each case, $E_{1-}$ and $E_{2-}$ are expressed in terms of the S-parameters and the illuminating electric field. Additionally, the assumption of plane wave illumination allows us to express the magnetic field quantities in terms of the electric field and wave impedance of each region. These four equations are simplified and assembled into a matrix equation to express the surface parameters in terms of S-parameters.

$$\frac{1}{2}\begin{pmatrix} Y & \chi \\ -\gamma & Z \end{pmatrix} = \begin{pmatrix} \frac{1}{\eta_1} - \frac{S_{11}}{\eta_1} - \frac{S_{21}}{\eta_2} & \frac{1}{\eta_2} - \frac{S_{22}}{\eta_2} - \frac{S_{12}}{\eta_1} \\ 1 + S_{11} - S_{21} & -1 - S_{22} + S_{12} \end{pmatrix} \begin{pmatrix} 1 + S_{11} + S_{21} & 1 + S_{22} + S_{12} \\ \frac{1}{\eta_1} - \frac{S_{11}}{\eta_1} + \frac{S_{21}}{\eta_2} & -\frac{1}{\eta_2} + \frac{S_{22}}{\eta_2} - \frac{S_{12}}{\eta_1} \end{pmatrix}^{-1} \tag{8}$$

The form of Equation (8) is convenient for calculating the surface parameters that will implement the desired S-parameters. However, re-arranging Equation (8) and simplifying to solve the S-parameter quantities provides

$$S_{11} = \frac{1}{\sigma}\left(-Y + Z\frac{1}{\eta_1\eta_2} + \left[\frac{1}{4\eta_1}[(2-\gamma)(2-\chi) + YZ] - \frac{1}{4\eta_2}[(2+\gamma)(2+\chi) + YZ]\right]\right) \tag{9}$$

$$S_{12} = \frac{1}{\sigma}\left(\frac{1}{2\eta_2}[(2-\gamma)(2+\chi) - YZ]\right) \tag{10}$$

$$S_{21} = \frac{1}{\sigma}\left(\frac{1}{2\eta_1}[(2+\gamma)(2-\chi) - YZ]\right) \tag{11}$$

$$S_{22} = \frac{1}{\sigma}\left(-Y + Z\frac{1}{\eta_1\eta_2} - \left[\frac{1}{4\eta_1}[(2-\gamma)(2-\chi) + YZ] - \frac{1}{4\eta_2}[(2+\gamma)(2+\chi) + YZ]\right]\right) \tag{12}$$

$$\sigma = Y + Z\frac{1}{\eta_1\eta_2} + \left[\frac{1}{4\eta_1}[(2-\gamma)(2-\chi) + YZ] + \frac{1}{4\eta_2}[(2+\gamma)(2+\chi) + YZ]\right] \tag{13}$$

In the case of a lossless metasurface, the surface parameters $Y = jB$ and $Z = jX$ are purely imaginary, while $\gamma = \chi = R$ are real quantities [32]. In this case, Equations (9)–(12) can be further simplified, as shown in Equations (14)–(17). Note that $S_{21} = S_{12}$ only when $\eta_1 = \eta_2$, and $S_{11} = S_{22}$ only when $R = 0$ and $\eta_1 = \eta_2$.

$$S_{11} = \frac{1}{\sigma}\left( j\left[\frac{X}{\eta_1\eta_2} - B\right] + \left[\frac{1}{4\eta_1}\left[(2-R)^2 - BX\right] - \frac{1}{4\eta_2}\left[(2+R)^2 - BX\right]\right]\right) \tag{14}$$

$$S_{12} = \frac{1}{\sigma}\left(\frac{1}{2\eta_2}\left[4 - R^2 + BX\right]\right) \tag{15}$$

$$S_{21} = \frac{1}{\sigma}\left(\frac{1}{2\eta_1}\left[4 - R^2 + BX\right]\right) \tag{16}$$

$$S_{22} = \frac{1}{\sigma}\left( j\left[\frac{X}{\eta_1\eta_2} - B\right] - \left[\frac{1}{4\eta_1}\left[(2-R)^2 - BX\right] - \frac{1}{4\eta_2}\left[(2+R)^2 - BX\right]\right]\right) \tag{17}$$

$$\sigma = j\left[\frac{X}{\eta_1\eta_2} + B\right] + \left[\frac{1}{4\eta_1}\left[(2-R)^2 - BX\right] + \frac{1}{4\eta_2}\left[(2+R)^2 - BX\right]\right] \tag{18}$$

We can also determine the limitations that are placed on the S-parameters when passive, lossless, and reciprocal restrictions are enforced. For a bi-isotropic metasurface, the S-parameters represent a two-port network, as described in Equation (5). Each element is a complex number, so there are eight total variables (four real and four imaginary quantities). For a reciprocal network, $S_{21} = S_{12}$ when the port impedances are the same. This relationship shows that both of the transmission coefficients are the same in amplitude and phase. However, a different relationship is needed for the case of the bi-isotropic metasurface, since the port impedances are different. Reciprocity is satisfied when the transmission phase shift and transmitted power are the same for each direction of illumination. When the port impedances are not equal, the electric field amplitude will change, depending on the wave impedance of the medium in order to satisfy the reciprocity conditions, so $|S_{21}| \neq |S_{12}|$.

In order to determine the reciprocity relationship for a bi-isotropic metasurface, consider two cases: (i) where the metasurface is only illuminated from region 1 and transmitted power is determined in region 2, and (ii) the metasurface is only illuminated from region 2 and transmission measured in region 1. By equating the transmitted power in both cases, we arrive at

$$\frac{\eta_1}{\eta_2}|S_{21}|^2 = \frac{\eta_2}{\eta_1}|S_{12}|^2 \tag{19}$$

While Equation (19) provides a relationship between the transmission coefficient magnitudes, reciprocity also requires that the transmission phase be the same. Applying this and assuming the wave impedance of each region is real, we arrive at

$$\sqrt{\frac{\eta_1}{\eta_2}}S_{21} = \sqrt{\frac{\eta_2}{\eta_1}}S_{12}. \tag{20}$$

Note that Equations (15) and (16) satisfy this relationship, since the metasurface parameters were restricted to be reciprocal.

In order to enforce the lossless condition, the time-average power that is absorbed by the metasurface must be zero. This is calculated as

$$P_{avg} = \frac{1}{2}\left\{\left(\begin{bmatrix}E_{1+}\\E_{2+}\end{bmatrix} + \begin{bmatrix}E_{1-}\\E_{2-}\end{bmatrix}\right)^T\left(\begin{bmatrix}H_{1+}\\H_{2+}\end{bmatrix} - \begin{bmatrix}H_{1-}\\H_{2-}\end{bmatrix}\right)^*\right\} = 0 \tag{21}$$

By applying the plane wave relation between the electric and magnetic fields, and expressing $E_{1-}$ and $E_{2-}$ in terms of the S-parameters from Equation (5), Equation (21) becomes

$$P_{avg} = \frac{1}{2}Re\left\{\left([E_+] + [S][E_+]\right)^T\left([1/\eta][E_+] - [1/\eta][S][E_+]\right)^*\right\} = 0 \tag{22}$$

where

$$[1/\eta] = \begin{bmatrix} 1/\eta_1 & 0 \\ 0 & 1/\eta_2 \end{bmatrix}, \qquad [E_+] = \begin{bmatrix} E_{1+} \\ E_{2+} \end{bmatrix}. \tag{23}$$

Simplifying Equation (22) and utilizing the reciprocity relationship in (20) results in three equations that must be satisfied in order to implement a lossless and reciprocal metasurface.

$$1 = |S_{11}|^2 + \frac{\eta_1}{\eta_2}|S_{21}|^2 \tag{24}$$

$$1 = |S_{22}|^2 + \frac{\eta_1}{\eta_2}|S_{21}|^2 \tag{25}$$

$$0 = |S_{11}|\cos(\phi_{S11} - \phi_{S21} + \phi_{E1+} - \phi_{E2+}) + |S_{22}|\cos(\phi_{S21} - \phi_{S22} + \phi_{E1+} - \phi_{E2+}) \tag{26}$$

These three equations under-define the six independent scattering matrix variables. Consequently, three variables can be chosen freely without violating the lossless and reciprocal conditions. Specifically, Equations (24) and (25) provide the ability to choose one amplitude of the scattering matrix. If $|S_{21}|$ is chosen, as is commonly the case, then $|S_{11}| = |S_{22}|$, and a phase constraint is obtained from Equation (26)

$$\phi_{S11} - \phi_{S21} = \phi_{S21} - \phi_{S22} + \pi \tag{27}$$

where two phase shifts of the S-parameters can be freely chosen.

Therefore, for a bi-isotropic metasurface to be both lossless and reciprocal, three degrees of freedom exist in its S-parameters: one S-parameter amplitude and two S-parameter phases. These three degrees of freedom are set through the design choices of the metasurface. It is worth recalling that enforcing lossless and reciprocal behavior in the surface parameters for the bi-isotropic metasurface results in $Y = jB$, $Z = jX$, and $\gamma = \chi = R$. Thus, three distinct surface parameters can be chosen to achieve three desired scattering properties.

### 3. Bi-Isotropic Metasurfaces: Bandwidth and Quality Factor

In practice, bi-isotropic metasurfaces typically rely on resonant structures to produce the strong field interactions that are required to perform the desired field transformations. However, the use of resonances places inherent limitations on the bandwidth. In this section, the relationship between matching networks and bi-isotropic metasurfaces is considered, and the quality factor of a metasurface realized using three impedance sheets is defined. We demonstrate that the quality factor can be used as a metric to predict the metasurface's bandwidth and identify unit cells that degrade the performance of inhomogeneous metasurfaces.

We consider the following example to understand the relationship between impedance matching networks and bi-isotropic metasurfaces. Suppose that there is a planar interface between air and alumina ($\varepsilon_r = 9.4$), as in Figure 2, and the goal is to maximize the power that is transferred across the interface. Because the intrinsic wave impedances of the media are real, this amounts to minimizing the amplitude of the reflected wave. To do this, the input impedance of the metasurface must be equal to the wave impedance of the incident wave, $Z_{in}$. Because the two media have different wave impedances the metasurface must transform the wave impedance of the transmitted wave, $Z_L$, to that of the incident wave, $Z_{in}$. In this scenario, the metasurface acts as an impedance matching layer. Here, the impedance matching layer is analogous to an impedance matching network from circuit theory, like an L or T-network, as shown in Figure 3. From circuit theory, it is known that a complex load impedance can be matched to a complex source impedance using either an L-, T-, or $\pi$-network [33]. The L-network contains two degrees of freedom allowing for the real and imaginary components of the input impedance to be matched. For an L-network, the solution is unique (all the degrees of freedom are used) and no other characteristics of the impedance match, such as its bandwidth or the transmission phase, can be controlled.

Adding a third degree of freedom to the L-network produces a T- or $\pi$-network. This additional degree of freedom can be used to control the bandwidth or the transmission phase. Bi-isotropic metasurfaces are like T-matching networks for fields [34]. They have three degrees of freedom that allow for impedance matching with phase or bandwidth control [35]. To illustrate this idea, we consider a metasurface that impedance matches a normally incident plane wave on an air-alumina ($\varepsilon_r = 9.4$) interface over a maximum bandwidth, as shown in Figure 2.

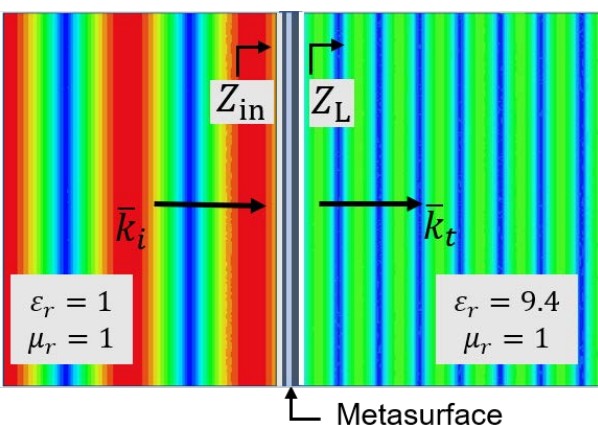

**Figure 2.** A metasurface at the interface between air and alumina half-spaces. The metasurface is used to impedance match a normally incident plane wave traveling from the region of air into the alumina.

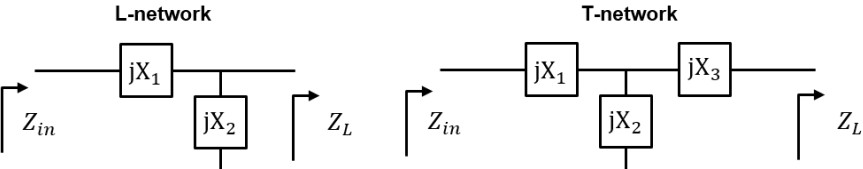

**Figure 3.** L and T circuit network topologies used for impedance matching in circuit theory.

In order to design the impedance matching metasurface, recall that a bi-isotropic metasurface can be viewed as a two-port network that controls one scattering amplitude and two scattering phases. Therefore, designing a lossless, reflectionless, and bi-isotropic metasurface is equivalent to designing a lossless two-port impedance matrix (Z-matrix) that impedance matches a load impedance $Z_L = |Z_L|e^{j\phi_L}$ to a source impedance $Z_i = |Z_{in}|e^{j\phi_{in}}$ with an arbitrary transmission phase $\phi_{21}$ [36]. Consider a general lossless two-port Z-matrix in order to determine the Z-matrix that provides the desired functionality,

$$\begin{pmatrix} V_1 \\ V_2 \end{pmatrix} = j \begin{pmatrix} X_{11} & X_{12} \\ X_{21} & X_{22} \end{pmatrix} \begin{pmatrix} I_1 \\ I_2 \end{pmatrix} \tag{28}$$

Imposing the impedance boundary conditions and enforcing power conservation on (28) produces the following system of equations,

$$\begin{pmatrix} 1 \\ r_v e^{j\phi_{21}} \end{pmatrix} = j \begin{pmatrix} X_{11} & X_{12} \\ X_{21} & X_{22} \end{pmatrix} \begin{pmatrix} 1 \\ -r_v e^{j\phi_{21}} \end{pmatrix} \tag{29}$$

where $r_v^2 = \frac{Z_L}{Z_{in}} \left| \frac{\cos \phi_{in}}{\cos \phi_L} \right|$ and $\phi_{21} = \angle V_2 - \angle V_1$. Splitting (29) into its real and imaginary components allows for the elements of the Z-matrix to be solved in terms of $Z_L$, $Z_{in}$ and $\phi_{21}$,

$$\begin{pmatrix} X_{11} & X_{12} \\ X_{21} & X_{22} \end{pmatrix} = \begin{pmatrix} |Z_{in}| \cos(\phi_{21} - \phi_L) & |Z_{in}| r_v \cos(\phi_L) \\ |Z_{in}| r_v \cos(\phi_L) & |Z_L| \cos(\phi_{21} + \phi_{in}) \end{pmatrix} \csc(\phi_{21} + \phi_{in} - \phi_L) \tag{30}$$

From (30), it is clear that the required two-port network is reciprocal, since $X_{12} = X_{21}$, and it has three degrees of freedom.

Three cascaded sheet impedances, as shown in Figure 4, can be used to realize a metasurface with a Z-matrix given by (30), as in [5]. Expressing Figure 4 in terms of its Z-matrix, and solving for the necessary impedance sheets, results in the following expressions for the sheets in terms of the elements of (30),

$$Z_{s1} = -j\frac{Z_0 \sin(\beta d)}{\cos(\beta d) + (\frac{X_{12} + X_{22}}{\det Z})Z_0 \sin(\beta d)} \tag{31}$$

$$Z_{s2} = -j\frac{Z_0^2 \sin^2(\beta d) X_{12}}{\det Z + X_{12} Z_0 \sin(2\beta d)} \tag{32}$$

$$Z_{s3} = -j\frac{Z_0 \sin(\beta d)}{\cos(\beta d) + (\frac{X_{12} + X_{11}}{\det Z})Z_0 \sin(\beta d)}, \tag{33}$$

where det Z is the determinant of the Z-matrix and $\beta$ and $Z_0$ are the wavenumber and wave impedance of the dielectric spacers, respectively. Once the input and load impedances, spacer thickness, and the transmission phase are specified, (30)–(33) can be used to determine the necessary impedance sheets to implement the metasurface.

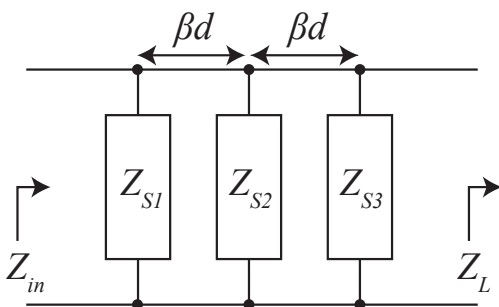

**Figure 4.** Bi-isotropic metasurface realized using three impedance sheets that are separated by dielectric spacers with thickness d.

In order to maximize the bandwidth of the impedance match, a method is needed for comparing the metasurface's bandwidth for different transmission phases. Here, an expression for the metasurface's quality factor as a function of the transmission phase is derived for this purpose. The quality factor of a three-sheet metasurface is defined as,

$$Q = \omega_0 \frac{2W_e}{P_d}, \tag{34}$$

where $\omega_0$ is the angular resonant frequency, $W_e$ is the average electric energy stored in the network at $\omega_0$, and $P_d$ is the power dissipated in the network. In order to calculate the quality factor using (34), the impedance sheets (31)–(33) are expressed in terms of lumped capacitances and inductances. The dielectric spacers in the metasurface are assumed to be electrically thin, so they can be modeled as lumped $\pi$-networks. Therefore, if the dielectric spacers are electrically thin and the source and load impedances are purely real, then the quality factor of the metasurface can be expressed as

$$Q = \frac{\omega_0}{2}\left(Z_{in}(C_{s1} + \frac{\beta d}{2\omega_0 Z_0}) + R_{int}(C_{s2} + \frac{\beta d}{\omega_0 Z_0}) + Z_L(C_{s3} + \frac{\beta d}{2\omega_0 Z_0})\right), \tag{35}$$

where, $R_{int} = \frac{Z_{in} + Z_L + \sqrt{Z_{in} Z_L} \cos \phi_{21}}{\sin^2 \phi_{21}} \frac{(Z_0 \sin \beta d)^2}{Z_{in} Z_L}$, and $C_{si}$ is the capacitance of the $i$th impedance sheet (if the sheet is inductive, then $C_{si} = 0$). If the load impedances are not purely real, then the imaginary part of the load can be absorbed into either $Z_{s1}$ or $Z_{s3}$, and (35) can

still be used. The quality factor, $Q$, of the metasurface will be used to approximate the fractional bandwidth, $FBW = BW/f_0$, where $BW$ is the 3 dB bandwidth of each unit cell. However, due to the presence of multiple resonances this approximation is only valid when the resonances are well separated in frequency.

The quality factor expression (35) can be used to maximize the bandwidth of an impedance matching bi-isotropic metasurface. For a normally incident plane wave, the relevant impedance is $Z_{in}$ = 377 Ω. Let us assume that $Z_L$ = 123 Ω, and the spacers are free-space with a thickness $d = \lambda_0/20$. Using (35) to calculate the quality factor and the fractional bandwidth versus transmission phase produces Figure 5. Figure 5a plots the quality factor which is minimized at a transmission phase of $\phi_{21} = -68.5°$. Figure 5b shows that this transmission phase is predicted to maximize the bandwidth, and Figure 5c shows that it produces the maximum 3dB bandwidth. The metasurface with this transmission phase is composed of the following impedance sheets: $Z_{s1} = 1/(j\omega C_{s1}) = -j468.9$ Ω, $Z_{s2} = 1/(j\omega C_{s2}) = -j641.9$ Ω, and $Z_{s3} = j\omega L_{s3} = j38.5$ kΩ. The metasurface performance is simulated in Ansys HFSS while using dispersive impedance sheets that correspond to the following lumped elements: $C_{s1}$ = 33.9 *fF*, $C_{s2}$ = 24.8 *fF*, and $L_{s3}$ = 612.7 *nH*. Figure 6 shows the transmission magnitudes from this simulation, where they are compared to a quarter-wave transformer and the bare interface without any impedance matching. The metasurface has a size and bandwidth comparable to a quarter-wave transformer. However, it does not require the realization of a medium with the dielectric constant $\varepsilon_r = \sqrt{9.4}$, which can be heavy and challenging to manufacture.

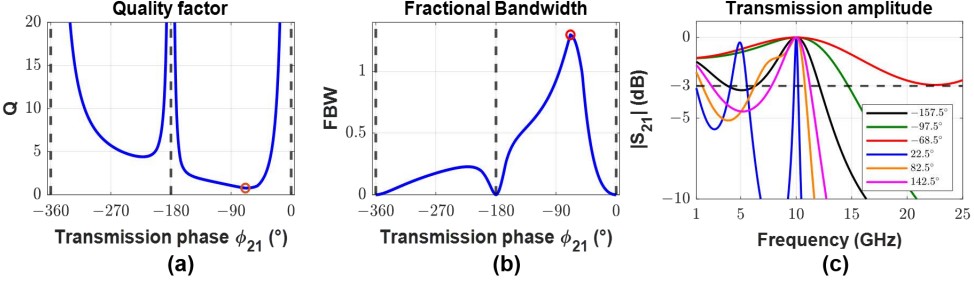

**Figure 5.** The quality factor, fractional bandwidth, and the magnitude of the frequency response for metasurfaces that provide impedance matching with six different transmission phases. (**a**) The quality factor is minimized when the transmission phase is $-68.5°$ (red circle). (**b**) The fractional bandwidth is maximized at $-68.5°$ (red circle). (**c**) Plots of the transmission amplitude over frequency for several transmission phases and the maximum bandwidth is observed when the transmission phase is $-68.5°$, as predicted by the quality factor.

In practice, a metasurface's impedance sheets are typically realized using subwavelength metal or dielectric patterning on support structures. Here, we will consider the impact this has on the bandwidth of the metasurface. First, we will consider the effect of using subwavelength patterned sheets. Subwavelength unit cells that are non-resonant exhibit a response of either a capacitive or inductive sheet impedance [37,38]. This indicates that modeling the patterned sheets as impedance sheets should not result in a significant bandwidth reduction when the metasurface is realized in practice. However, it may be necessary to modify the design to include additional impedance sheets to avoid extreme impedance values that are difficult to realize in practice. If impedance sheets cannot be realized at the design frequency due to manufacturing difficulties, an alternative design approach may be required, such as using detuned resonant elements. Their responses will be more narrowband.

We will also consider the effect of using a dielectric spacer as the support structure. For a metasurface that is designed using a non-magnetic dielectric spacer with a relative permittivity $\varepsilon_r$, its quality factor is given by,

$$Q = \frac{\omega_0}{2}\left(Z_{in}(C_{s1} + \frac{\varepsilon_r\varepsilon_0 d}{2}) + R_{int}(C_{s2} + \varepsilon_r\varepsilon_0 d) + Z_L(C_{s3} + \frac{\varepsilon_r\varepsilon_0 d}{2})\right). \tag{36}$$

If the dielectric spacer is electrically thin, then the sheet capacitances can be approximated as,

$$C_{s1} = \frac{1}{\omega}\left(\frac{1}{\omega\mu_0 d} + \frac{X_{22} + X_{12}}{\det(Z)}\right) - \frac{\varepsilon_r\varepsilon_0 d}{2} \tag{37}$$

$$C_{s2} = \frac{1}{\omega}\left(\frac{2}{\omega\mu_0 d} + \frac{\det(Z)}{X_{12}}\frac{1}{(\omega\mu_0 d)^2}\right) - \varepsilon_r\varepsilon_0 d \tag{38}$$

$$C_{s3} = \frac{1}{\omega}\left(\frac{1}{\omega\mu_0 d} + \frac{X_{11} + X_{12}}{\det(Z)}\right) - \frac{\varepsilon_r\varepsilon_0 d}{2}, \tag{39}$$

when the sheet impedance $Z_{si}$ is capacitive (see Appendix A). Otherwise, the impedance sheet is inductive and it can be ignored in the calculation of the quality factor. Additionally, when the spacer is electrically thin, $R_{int}$ does not depend on the dielectric constant (see Appendix A). Accordingly, the only terms in (36) that depend on the permittivity are the capacitance terms $C_{si}$, $\frac{\varepsilon_r\varepsilon_0 d}{2}$, and $\varepsilon_r\varepsilon_0 d$. When considering the terms $C_{s1} + \frac{\varepsilon_r\varepsilon_0 d}{2}$, $C_{s2} + \varepsilon_r\varepsilon_0 d$, and $C_{s3} + \frac{\varepsilon_r\varepsilon_0 d}{2}$ individually with (37)–(39) in them, it becomes apparent that the quality factor is unaffected by the dielectric spacer for capacitive impedance sheets. However, if any of the impedance sheets are inductive, then the dielectric spacer will increase the quality factor, thereby reducing the metasurface's bandwidth.

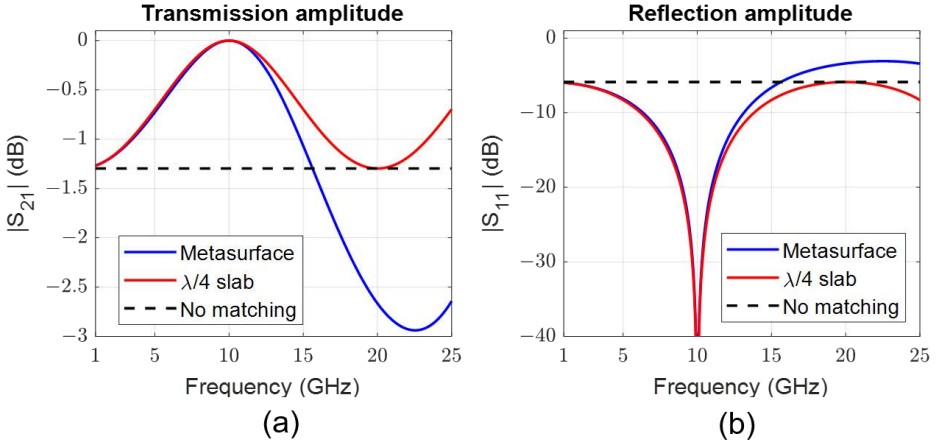

**Figure 6.** Plots of the (**a**) transmission and (**b**) reflection magnitudes for the interface with the metasurface ($\phi_{21} = -68.5°$), a quarter-wave transformer, and with no impedance matching (bare interface). The simulations of the metasurface were performed in Ansys HFSS. The metasurface has a bandwidth that is comparable to a quarter-wave transformer.

In addition to bandwidth information, the quality factor also provides information that can guide the design of inhomogeneous metasurfaces where local periodicity is assumed. Obtaining good performance from a metasurface that is designed assuming local periodicity requires that neighboring unit cells produce fields that are approximately the same, i.e., the fields vary smoothly along the surface without large discontinuities in the amplitude or phase. In this work, it has been found that the quality factor and its first derivative with respect to transmission phase can help the designer to select unit cells that satisfy the assumption of local periodicity.

The quality factor, as given by (35), is divergent at transmission phases near $\phi_{21} = 0°, -180°$, and $-360°$, indicating that the unit cells that are required to achieve these transmission phases possess large quality factors. Large quality factors are associated with strong resonances that are sensitive to perturbations in the surrounding environment and are lossy when realized in practice. Therefore, these unit cells should be avoided. Additionally, areas where (35) is not smooth (i.e., points where the first derivative is discontinuous or undefined) indicate transmission phases where the reactance of at least one of the impedance sheets changes sign. These points should also be avoided because they identify transmission phases where the required reactance values display asymptotic behavior. This introduces rapid variations in the values of the impedance sheets and fields in the metasurface that invalidate the assumption of local periodicity.

In order to see how this information can be used, consider a metasurface embedded in free-space that refracts a normally incident plane wave to $70°$ at a frequency of $f_0 = 10$ GHz. This requires a gradient metasurface: an inhomogeneous metasurface that imposes a linear phase gradient on an impinging wave-front to produce reflection or refraction in a desired direction [39]. Refraction requires the metasurface to alter the transverse wavenumber of an incident plane wave ($k_i = k \sin(\theta_i)$) to produce the desired refracted wavenumber ($k_t = k \sin(\theta_t)$), where $k$ is the wavenumber in the surrounding medium. Therefore, the metasurface must impart transverse momentum that is equal to $\Delta k = k_t - k_i$. Practically, this is realized by discretizing the metasurface into $N$ sub-wavelength unit cells that are of size $D = \frac{2\pi}{N \max(k_i, k_t)}$, each possessing a transmission phase $\phi_j$, such that $\Delta\phi = \phi_{j+1} - \phi_j = -\Delta k D$. Each unit cell must be reflectionless in order to maximize the transmitted power into the refracted wave. This means that impedance matching and phase control are required, so (31)–(33) can be used to design the unit cells of the metasurface.

For this example, the metasurface will have 10 unit cells per transverse wavelength (in free-space) and the spacers will be assumed to be free-space with a thickness $d = \lambda_0/25$. As a first pass at the design, the metasurface is designed to impose a linear phase gradient with the unit cell transmission phases that are shown in Table 1. The required sheet impedances, as shown in Figure 7b, are solved using (31)–(33) and one period (10 unit cells) of the metasurface is simulated in COMSOL using periodic boundary conditions. Figure 7c shows the results.

**Table 1.** The unit cell transmission phases ($\phi_{21}$) used in the design of the gradient metasurface for plane wave refraction. The original phase gradient corresponds to the linear phase gradient. The perturbed phase gradient corresponds to the adjusted phases used to improve the performance of the metasurface.

| Unit Cell | $\phi_{21}$ (Original) | $\phi_{21}$ (Perturbed) |
|:---:|:---:|:---:|
| 1 | $-18°$ | $-31°$ |
| 2 | $-54°$ | $-54°$ |
| 3 | $-90°$ | $-90°$ |
| 4 | $-126°$ | $-126°$ |
| 5 | $-162°$ | $-147°$ |
| 6 | $-198°$ | $-216°$ |
| 7 | $-234°$ | $-234°$ |
| 8 | $-270°$ | $-270°$ |
| 9 | $-306°$ | $-306°$ |
| 10 | $-342°$ | $-330°$ |

The metasurface that is designed using this phase gradient exhibits significant reflections and the transmitted wave is not purely refracted. A slight perturbation of the linear phase gradient can be used to improve the performance. The appropriate perturbed phase gradient is found using the quality factor and its first derivative with respect to transmission phase. To find the problematic transmission phases in the original design, plots of the quality factor and its first derivative are shown in Figure 8. By inspecting the plots, four unit cells with problematic transmission phases are identified: 1, 5, 6, and

10. Unit cells 1, 5, and 6 are problematic, because they are near points where (35) is not smooth, and unit cell 10 is problematic due to its large quality factor. The problematic transmission phases are adjusted to improve the performance of the metasurface, as shown in Table 1 and Figure 9. These phase shifts reduce the maximum unit cell quality factor by approximately 10 and force the reactance of each impedance sheet to change sign only once at $\phi_{21} = -180°$.

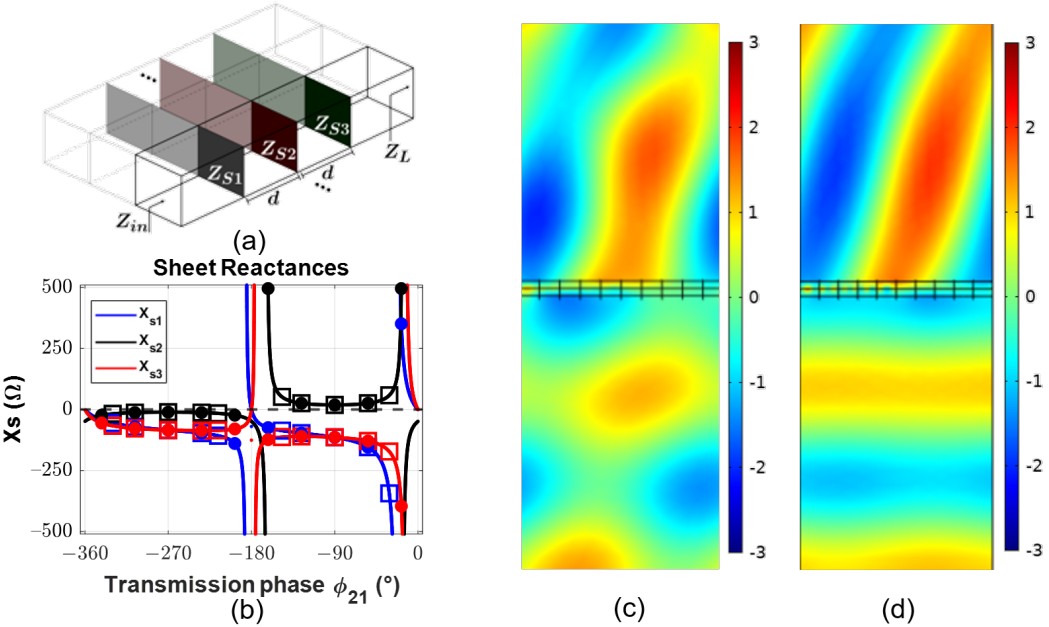

**Figure 7.** (**a**) The depiction of an inhomogeneous, bi-isotropic metasurface implemented as a three-sheet cascade in free-space. (**b**) Plots of the sheet reactances for different transmission phases. The solid circles indicate the values used for the linear phase gradient and the empty squares indicate the sheet values used for the perturbed phase gradient. (**c**) Full-wave simulation results for the real part of the electric field using the metasurface with a linear phase gradient. (**d**) Full-wave simulation results for the real part of the electric field using the metasurface with a perturbed phase gradient.

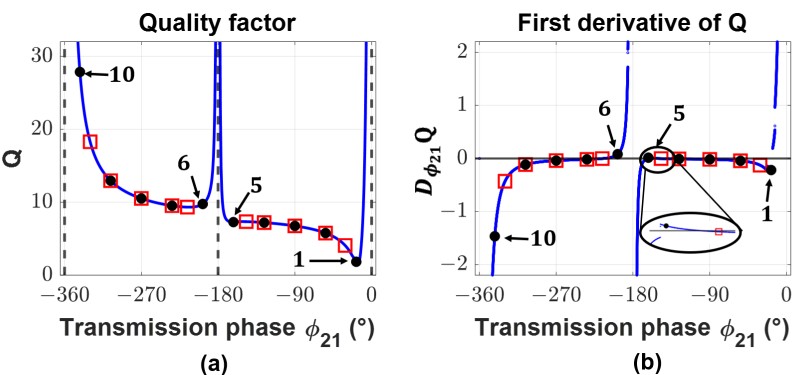

**Figure 8.** (**a**) The quality factor of the metasurface unit cells versus transmission phase. (**b**) The first derivative of the quality factor with respect to the transmission phase. The solid black circles indicate the values corresponding to the linear phase gradient and the hollow red squares indicate the adjusted values used for the perturbed phase gradient.

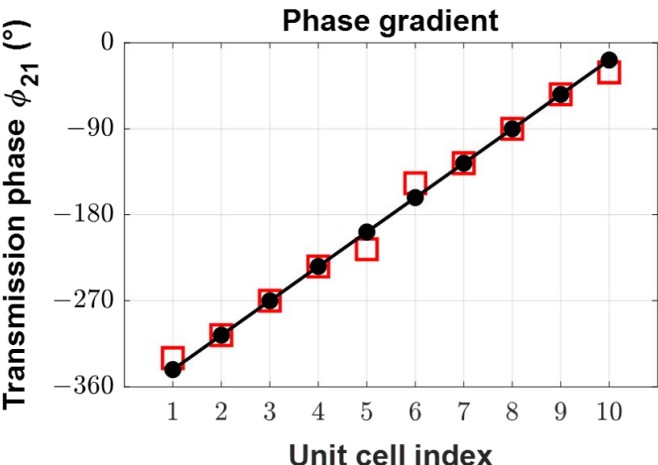

**Figure 9.** A comparison of the transmission phases used for the original and perturbed phase gradients. The solid black circles indicate the transmission phases used for the linear phase gradient. The empty red squares indicate the transmission phases that are used for the perturbed phase gradient.

The metasurface is redesigned with the modified transmission phases and Figure 7b shows the required sheet impedances. Ten unit cells of the metasurface are again simulated in COMSOL using periodic boundary conditions, and the results are shown in Figure 7d. We see that the redesigned metasurface performs significantly better than the analytical design. This indicates that avoiding transmission phases that require a large quality factor or exist near non-smooth or asymptotic regions of $Q(\phi_{21})$ can improve the performance of gradient metasurfaces that are designed using the local periodicity assumption.

Violations of local periodicity (like those discussed above) can present challenges when realizing inhomogeneous metasurfaces where local periodicity has been assumed. Issues that arise from these violations have been implicitly handled in the literature in a variety of ways. In [7], the phase gradient was altered to improve the metasurface's performance by reducing the transmission losses. On the other hand [36,40,41], made the spacers between the sheets extremely thin $d < \lambda/40$. This generally increases the quality factor of the unit cells, but it has the benefit of shifting the transmission phases where all three impedance sheets transition from capacitive to inductive to occur at the same point. This means that shrinking the spacings makes it easier to select transmission phases that avoid regions where (35) is not smooth. Consequently, extremely thin spacings can improve the design performance at the expense of increasing manufacturing difficulties and producing higher quality factors: lower bandwidths. Alternatively, PEC [42] or PMC [29] baffles have been used to eliminate inter-cell coupling to validate the assumption of local periodicity. However, in practice, the use of PEC baffles presents a manufacturing challenge and PMC baffles cannot be realized. These examples indicate a design trade-off between manufacturability and performance when realizing inhomogeneous metasurfaces. Using the quality factor, as shown in this section, provides an alternative way to improve design performance. It can be used to systematically identify the problematic unit cells and adjust them where possible to allow for the trade-off between performance and manufacturability to be balanced. An alternative to this approach is to avoid the assumption of local periodicity and model interactions between unique unit cells through homogenization and integral equations, as reported in [43].

## 4. Scattering from Bianisotropic Metasurfaces

While the scattering of plane waves was analyzed for the simplified case of a biisotropic metasurface shown in Section 2, it is worthwhile to consider the general case where isotropy is not assumed. Following the general process of Section 2, the boundary conditions of Equation (2) can be expressed in terms of the S-parameters (where each S-

parameter term is the $2 \times 2$ matrix of Equation (1)). Note that $\hat{z} \times \begin{bmatrix} E_x & E_y \end{bmatrix}^T = \boldsymbol{n} \begin{bmatrix} E_x & E_y \end{bmatrix}^T$. Expressing the surface parameters in terms of the S-parameters gives [5]

$$
\frac{1}{2}\begin{pmatrix} \boldsymbol{Y} & \boldsymbol{\chi} \\ \boldsymbol{\gamma} & \boldsymbol{Z} \end{pmatrix} = \begin{pmatrix} \frac{\boldsymbol{I}}{\eta_1} - \frac{\boldsymbol{S_{11}}}{\eta_1} - \frac{\boldsymbol{S_{21}}}{\eta_2} & \frac{\boldsymbol{I}}{\eta_2} - \frac{\boldsymbol{S_{22}}}{\eta_2} - \frac{\boldsymbol{S_{12}}}{\eta_1} \\ \boldsymbol{n}(\boldsymbol{I} + \boldsymbol{S_{11}} - \boldsymbol{S_{21}}) & -\boldsymbol{n}(\boldsymbol{I} + \boldsymbol{S_{22}} - \boldsymbol{S_{12}}) \end{pmatrix} \begin{pmatrix} \boldsymbol{I} + \boldsymbol{S_{11}} + \boldsymbol{S_{21}} & \boldsymbol{I} + \boldsymbol{S_{22}} + \boldsymbol{S_{12}} \\ \boldsymbol{n}\left(\frac{\boldsymbol{I}}{\eta_1} - \frac{\boldsymbol{S_{11}}}{\eta_1} + \frac{\boldsymbol{S_{21}}}{\eta_2}\right) & -\boldsymbol{n}\left(\frac{\boldsymbol{I}}{\eta_2} - \frac{\boldsymbol{S_{22}}}{\eta_2} + \frac{\boldsymbol{S_{12}}}{\eta_1}\right) \end{pmatrix}^{-1} \tag{40}
$$

Equation (40) can also be re-arranged so that the S-parameters are expressed in terms of surface parameters.

$$
\begin{pmatrix} \boldsymbol{S_{11}} & \boldsymbol{S_{12}} \\ \boldsymbol{S_{21}} & \boldsymbol{S_{22}} \end{pmatrix} = \begin{pmatrix} \frac{\boldsymbol{I}}{\eta_1} + \frac{\boldsymbol{Y}}{2} - \frac{\boldsymbol{\chi n}}{2\eta_1} & \frac{\boldsymbol{I}}{\eta_2} + \frac{\boldsymbol{Y}}{2} + \frac{\boldsymbol{\chi n}}{2\eta_2} \\ -\boldsymbol{n} + \frac{\boldsymbol{\gamma}}{2} - \frac{\boldsymbol{Z n}}{2\eta_1} & \boldsymbol{n} + \frac{\boldsymbol{\gamma}}{2} + \frac{\boldsymbol{Z n}}{2\eta_2} \end{pmatrix}^{-1} \begin{pmatrix} \frac{\boldsymbol{I}}{\eta_1} - \frac{\boldsymbol{Y}}{2} - \frac{\boldsymbol{\chi n}}{2\eta_1} & \frac{\boldsymbol{I}}{\eta_2} - \frac{\boldsymbol{Y}}{2} + \frac{\boldsymbol{\chi n}}{2\eta_2} \\ \boldsymbol{n} - \frac{\boldsymbol{\gamma}}{2} - \frac{\boldsymbol{Z n}}{2\eta_1} & -\boldsymbol{n} - \frac{\boldsymbol{\gamma}}{2} + \frac{\boldsymbol{Z n}}{2\eta_2} \end{pmatrix} \tag{41}
$$

Analyzing the degrees of freedom helps to determine the number of surface parameters required to realize a specified S-matrix, as in the bi-isotropic case that is discussed in Section 2. In the bianisotropic case, both of the polarizations are taken into account, which leads to a $4 \times 4$ scattering matrix and 16 complex numbers as its entries. In most cases, reciprocity is desired for metasurfaces, which results in a symmetric S-matrix (assuming the port impedances are identical)

$$
\begin{pmatrix} \boldsymbol{S_{11}} & \boldsymbol{S_{12}} \\ \boldsymbol{S_{21}} & \boldsymbol{S_{22}} \end{pmatrix} = \begin{pmatrix} \boldsymbol{S_{11}} & \boldsymbol{S_{12}} \\ \boldsymbol{S_{21}} & \boldsymbol{S_{22}} \end{pmatrix}^T \tag{42}
$$

which indicates that only 10 out of the 16 entries are actually independent. Because each complex number contains its real part and imaginary part, there are 20 free variables in total under the reciprocal condition. The S-matrix also has to be unitary if we further require the metasurface to be lossless:

$$
\begin{pmatrix} \boldsymbol{S_{11}} & \boldsymbol{S_{12}} \\ \boldsymbol{S_{21}} & \boldsymbol{S_{22}} \end{pmatrix}^T \begin{pmatrix} \boldsymbol{S_{11}} & \boldsymbol{S_{12}} \\ \boldsymbol{S_{21}} & \boldsymbol{S_{22}} \end{pmatrix}^* = \begin{pmatrix} 1 & 0 & 0 & 0 \\ 0 & 1 & 0 & 0 \\ 0 & 0 & 1 & 0 \\ 0 & 0 & 0 & 1 \end{pmatrix}. \tag{43}
$$

By incorporating the reciprocal condition (42) into the lossless condition (43), one can expand (43) into 10 different equations, which impose 10 additional restrictions on the 20 free variables. Consequently, for a reciprocal and lossless bianisotropic metasurface, there are 10 degrees of freedom in total.

A similar conclusion can be drawn by considering the surface parameters. Recall that $\boldsymbol{Y} = \boldsymbol{Y}^T$, $\boldsymbol{\gamma} = -\boldsymbol{\chi}^T$, and $\boldsymbol{Z} = \boldsymbol{Z}^T$ for a reciprocal metasurface [32]. There are three free entries in both of the $\boldsymbol{Y}$ and $\boldsymbol{Z}$ matrices, and four free entries in the $\boldsymbol{\gamma}$ or $\boldsymbol{\chi}$ matrix. Moreover, for the metasurface to be lossless, $\boldsymbol{Y}$ and $\boldsymbol{Z}$ must have purely imaginary entries, while $\boldsymbol{\gamma}$ and $\boldsymbol{\chi}$ must have purely real ones [32]. Again, it can be seen that the total degrees of freedom of the system is 10.

In practice, several sheets are usually cascaded and separated by dielectric spacers to form bianisotropic metasurfaces. Typically, these sheets only possess electric responses that are characterized by admittance tensors $\boldsymbol{Y}$, since they can be readily realized using metallic patterns. For bi-isotropic metasurfaces, or in the case where only a single polarization is of concern, we know that three sheets are enough to realize a specified response. However, the situation becomes more complicated for bianisotropic metasurfaces. When both of the polarizations are involved, a single lossless, reciprocal electric sheet provides three degrees of freedom under lossless and reciprocal conditions, i.e., the imaginary numbers $Y_{xx}$, $Y_{yy}$ and $Y_{xy} = Y_{yx}$. Therefore, at most, four sheets are required to realize an arbitrary reciprocal and lossless bianisotropic metasurface with 10 degrees of freedom. Although many bianisotropic metasurfaces can be realized with only three electric sheets, there are some cases in which introducing a fourth sheet is necessary. Examples include the

polarization rotators in [5,23]. The fourth sheet not only provides the required degree of freedom, but also enhances the operational bandwidth.

A network analysis technique, known as the wave matrix approach, was adopted in [23] in order to synthesize a cascaded sheet design. Wave matrices relate the forward and backward propagating electric fields on one side of the scatterer to those on the other side. For an arbitrary scatterer that is shown in Figure 10a, the wave matrix $M$ is defined as:

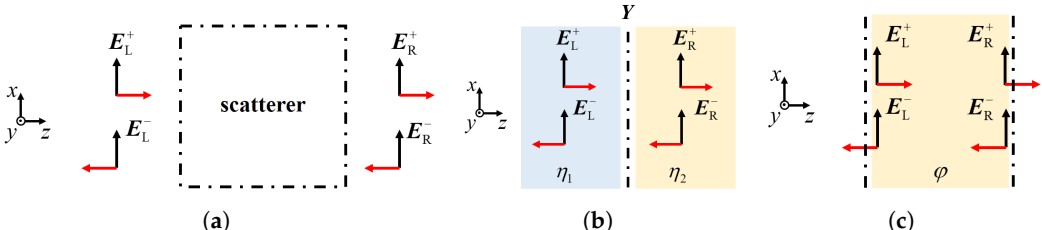

**Figure 10.** An illustration of the wave matrix and the constitutive blocks of the cascaded structure. (**a**) The definition of a wave matrix. (**b**) A metasurface interface between two dielectric media. (**c**) A dielectric spacer.

$$\begin{pmatrix} E_L^+ \\ E_L^- \end{pmatrix} = M \begin{pmatrix} E_R^+ \\ E_R^- \end{pmatrix}. \tag{44}$$

Similar to the S-matrices, wave matrices contain information regarding the incident and reflected waves. The advantage of using wave matrices is that they significantly simplify the analysis of cascaded structures, such as ABCD matrices. The wave matrix of a cascaded structure can be obtained by simply multiplying the wave matrices of its constitutive blocks. In our multi-layer metasurfaces, these blocks include metasurface interfaces across two dielectric media and dielectric spacers, as illustrated in Figure 10b,c, respectively. Their corresponding wave matrices can be derived from the boundary conditions, and they are explicitly shown in [23].

The procedure for synthesizing a reciprocal and lossless S-matrix is briefly outlined here. First, the desired S-matrix, $S_{\mathrm{spec}}$, is stipulated based on the required application. It is then converted to a wave matrix, as follows [23]:

$$M_{\mathrm{spec}} = \begin{pmatrix} I & 0 \\ S_{11,\mathrm{spec}} & S_{12,\mathrm{spec}} \end{pmatrix} \begin{pmatrix} S_{21,\mathrm{spec}} & S_{22,\mathrm{spec}} \\ 0 & I \end{pmatrix}^{-1}, \tag{45}$$

where $0$ represents a $2 \times 2$ null matrix. This wave matrix $M_{\mathrm{spec}}$ is known and it is set as the synthesis goal. It is worth noting that, if $S_{21,\mathrm{spec}}$ has a zero determinant, taking the inverse matrix in (45) becomes invalid. In this case, a perturbation can be introduced into $S_{21,\mathrm{spec}}$ to alleviate this problem. For simplicity, it is assumed that this S-matrix can be realized by cascading three sheets. Accordingly, Figure 11 displays the targeted structure and the cascaded wave matrix that relates $E_1^{\pm}$ to $E_4^{\pm}$ is

$$M_{\mathrm{casc}} = M_{\mathrm{sheet}}^{(1)} M_{\mathrm{dielectric}}^{(2)} M_{\mathrm{sheet}}^{(2)} M_{\mathrm{dielectric}}^{(3)} M_{\mathrm{sheet}}^{(3)} \tag{46}$$

in which $M_{\mathrm{sheet}}^{(1)}$, $M_{\mathrm{sheet}}^{(2)}$ and $M_{\mathrm{sheet}}^{(3)}$ are the admittance tensors that need to be solved. By setting $M_{\mathrm{spec}} = M_{\mathrm{casc}}$, and with some algebraic manipulation, one can find the admittance tensor of the second sheet $Y_2$ [23]:

$$e \otimes Y_2 = \frac{1}{A_2} \left( (e \otimes I) M_{\mathrm{spec}}(e \otimes I) - (e t_1 \Phi_2 t_2 \Phi_3 t_3 e) \otimes I \right). \tag{47}$$

The symbol $\otimes$ in (47) represents the Kronecker product of matrices and $A_2$ is some constant scalar. The matrix $t_1$ contains information regarding the dielectric interface where

the first sheet is located, $\Phi_2$ carries the phase information of the second dielectric spacer ($t_2$, $t_3$, and $\Phi_3$ are similarly defined), and $e$ is a constant matrix [23]:

$$t_1 = \frac{1}{2\eta_2}\begin{pmatrix} \eta_2 + \eta_1 & \eta_2 - \eta_1 \\ \eta_2 - \eta_1 & \eta_2 + \eta_1 \end{pmatrix}, \quad \Phi_2 = \begin{pmatrix} e^{j\varphi_2} & 0 \\ 0 & e^{-j\varphi_2} \end{pmatrix}, \quad e = \begin{pmatrix} 1 & 1 \\ -1 & -1 \end{pmatrix}. \tag{48}$$

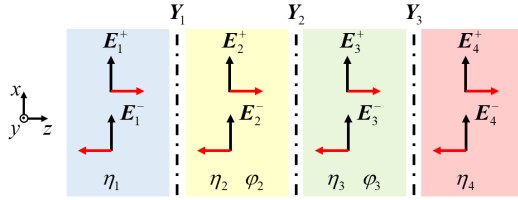

**Figure 11.** A cascaded metasurface structure consists of three sheets with only electric responses ($\gamma = \chi = Z = 0$).

Similarly, the admittance tensors of the first and third sheets $Y_1$ and $Y_3$ can be expressed in terms of $Y_2$,

$$
\begin{aligned}
e \otimes Y_1 &= \frac{1}{A_1}\left[ M_{\text{spec}}(e \otimes I) - (t_1\Phi_2 t_2 \Phi_3 t_3 e) \otimes I - \frac{\eta_2}{2}(t_1\Phi_2 e\Phi_3 t_3 e) \otimes Y_2 \right] \\
&\quad \cdot \left( I \otimes (I + \frac{B_1}{A_1}Y_2)^{-1} \right) \\
e \otimes Y_3 &= \frac{1}{A_3}\left( I \otimes (I + \frac{B_3}{A_3}Y_2)^{-1} \right) \\
&\quad \cdot \left[ (e \otimes I)M_{\text{spec}} - (et_1\Phi_2 t_2 \Phi_3 t_3) \otimes I - \frac{\eta_2}{2}(et_1\Phi_2 e\Phi_3 t_3) \otimes Y_2 \right]
\end{aligned}
\tag{49}
$$

where $A_1$, $B_1$, $A_3$, and $B_3$ are the constants explicitly calculated in [23]. A more complicated synthesis procedure involving four cascaded sheets is also discussed in [23], but the main idea follows the three-sheet case shown here.

## 5. Design Examples

In this section, several polarization-converting design examples are shown and discussed in order to illustrate the broad applicability of bianisotropic metasurfaces. For the details of the analysis and synthesis procedure, please refer to [23]. All of the devices considered here are realized by cascading several electric sheets and, thus, can be realized in practice using subwavelength patterned surfaces. The admittances of the electric sheets can be characterized analytically [37,38,44], or through full-wave extraction methods. However, evanescent coupling resulting from the fine features of the patterning may shift the unit cell's response when the impedance sheets are cascaded to produce the bianisotropic unit cells. In order to account for this, the impedance sheets will need to be designed, such that the desired response of the unit cell is maintained. Several examples of metasurfaces demonstrating the feasibility of this approach at microwave and millimeter wave frequencies can be found in [5,7,18,45].

### 5.1. Asymmetric Circular Polarizer

An asymmetric circular polarizer is the first device presented here. The device converts a right-hand circularly polarized incident wave to a left-hand circularly polarized transmitted wave. On the contrary, when the incident wave is left-hand circularly polarized, it is totally reflected. Figure 12a provides an illustration of the operation [23]. Accordingly, the device has S-parameters:

$$S_{12,\text{spec}} = S_{21,\text{spec}}^T = \frac{e^{j\phi}}{2}\begin{pmatrix} 1 & j \\ j & -1 \end{pmatrix}, \quad S_{11,\text{spec}} = S_{22,\text{spec}} = \frac{e^{j\phi}}{2}\begin{pmatrix} 1 & -j \\ -j & -1 \end{pmatrix}. \tag{50}$$

for normal incidence.

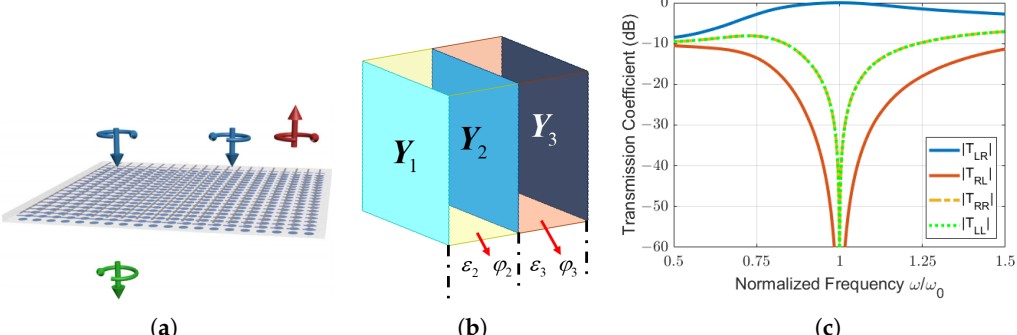

**Figure 12.** An asymmetric circular polarizer. (**a**) Polarization-converting operation of the metasurface [23]. (**b**) Unit cell of the asymmetric circular polarizer. (**c**) Simulated transmission coefficients.

Let us synthesize this asymmetric circular polarizer while using the wave matrix approach [23]. This specified S-matrix leads to a $S_{21,\text{spec}}$ with a vanishing determinant. Hence, a perturbation is required in this case, as noted in the previous section:

$$S_{12,\text{spec}} = S_{21,\text{spec}}^T = \frac{e^{j\phi}}{2} \begin{pmatrix} 1 & j \\ j & -e^{j1^\circ} \end{pmatrix}. \tag{51}$$

This device can be realized by cascading three sheets. Figure 12b shows a unit cell of the device. In this example, we stipulate both dielectric spacers to have a dielectric constant $\varepsilon_2 = \varepsilon_3 = 5$. It is assumed that the phase delay $\phi = 0$ and the electrical lengths of both dielectric spacers are $\varphi_2 = \varphi_3 = 2\pi/5$. By substituting these parameters into the design equations, (47) and (49), the following sheet admittance tensors are obtained,

$$Y_1 = \frac{j}{\eta_0} \begin{pmatrix} 0.73 & 1.00 \\ 1.00 & 0.72 \end{pmatrix}, \quad Y_2 = \frac{j}{\eta_0} \begin{pmatrix} 1268.31 & 5.52 \\ 5.52 & 1.43 \end{pmatrix}, \quad Y_3 = Y_1. \tag{52}$$

Synthesized admittance values (52) are modeled in Ansys HFSS as anisotropic boundary conditions for full-wave verification of the design. At frequencies other than the design frequency, $\omega_0$, the sheet admittances are assumed to obey Foster's reactance theorem. The eigenvalues of each sheet are first found by diagonalizing the tensors (52). We assume a capacitive frequency dependence if the resulting susceptances $B_0$ are positive,

$$B_c(\omega) = \frac{\omega}{\omega_0} B_0, \quad \text{if} \quad B_0 > 0. \tag{53}$$

On the other hand, negative susceptances are assumed to possess an inductive response

$$B_l(\omega) = \frac{\omega_0}{\omega} B_0, \quad \text{if} \quad B_0 < 0. \tag{54}$$

The assumptions, (53) and (54), are, in fact, reasonable since the admittances are realized by simple metallic patterns. Figure 12c displays the simulated frequency response of the asymmetric circular polarizer, where the transmission characteristics match the specified performance at the operating frequency (51).

*5.2. Polarization Rotator*

The second device considered in this section is a reflectionless polarization rotator that rotates the polarization of any linearly polarized incident wave by 90°, as shown in Figure 13a.

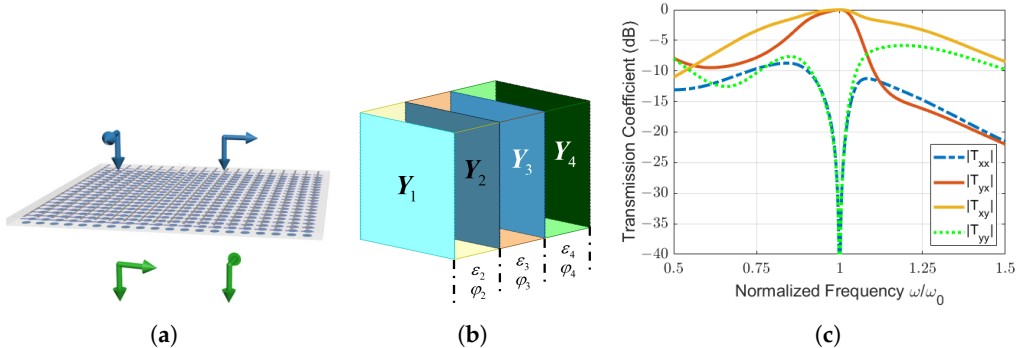

**Figure 13.** A linear polarization rotator. (**a**) Polarization-converting operation of the metasurface [23]. (**b**) Unit cell of the polarization rotator. (**c**) Simulated transmission coefficients.

The stipulated S-parameters for this example are,

$$S_{12,\text{spec}} = S_{21,\text{spec}}^T = e^{j\phi}\begin{pmatrix} 0 & 1 \\ -1 & 0 \end{pmatrix}, \quad S_{11,\text{spec}} = S_{22,\text{spec}} = e^{j\phi}\begin{pmatrix} 0 & 0 \\ 0 & 0 \end{pmatrix}. \tag{55}$$

As discussed earlier, four sheets are required to reconstruct the stipulated S-matrix. With the unit cell that is shown in Figure 13b, it is assumed that $\phi = \pi/4.5$, $\varepsilon_2 = \varepsilon_3 = \varepsilon_4 = 3.5$, and $\varphi_2 = \varphi_3 = \varphi_4 = \pi/5$. Following the synthesis procedure that is outlined in [23], the admittance tensors of each sheet are calculated to be:

$$Y_1 = \frac{j}{\eta_0}\begin{pmatrix} 5.01 & 0.77 \\ 0.77 & 0.13 \end{pmatrix}, \quad Y_2 = \frac{j}{\eta_0}\begin{pmatrix} 9.30 & 0 \\ 0 & 1.00 \end{pmatrix}$$

$$Y_3 = \frac{j}{\eta_0}\begin{pmatrix} 7.59 & -7.77 \\ -7.77 & 2.71 \end{pmatrix}, \quad Y_4 = \frac{j}{\eta_0}\begin{pmatrix} 2.57 & -1.30 \\ -1.30 & 2.57 \end{pmatrix}. \tag{56}$$

The metasurface unit cell was simulated in Ansys HFSS, and Figure 13c plots the resulting frequency response. Again, the transmission characteristics match the specified performance at the design frequency.

## 6. Conclusions

In this paper, two design procedures for realizing reciprocal bi-isotropic and bianisotropic metasurfaces using cascaded impedance sheets were reviewed. The design procedures use generalized sheet transition conditions (GSTCs) to relate bianisotropic surface parameters to the scattering parameters of the cascaded sheet impedances. Such approaches allow for metasurfaces with arbitrary lossless, reciprocal, and bianisotropic responses to be realized in practice. These design methods were then used to realize several examples of practical devices with phase and polarization control. Specifically, they were used to realize an asymmetric circular polarizer and a reflectionless, polarization rotator.

In addition to these design procedures, the quality factor for metasurfaces that are composed of three impedance sheets was introduced. The quality factor was shown to predict the bandwidth of a homogeneous metasurface. This was demonstrated through the design of an impedance matching metasurface with maximal bandwidth. It was also shown that the quality factor could be used to improve the performance of inhomogeneous metasurfaces. This was demonstrated through the design of a gradient metasurface for plane wave refraction. The unit cell quality factor was used to identify cells that degraded the overall metasurface performance, and it was used to select alternative unit cells that improved the overall performance. Such an approach can be used to balance manufacturability and performance trade-offs for metasurface devices.

**Author Contributions:** Conceptualization, A.G.; methodology, L.S., C.-W.L. and B.O.R.; software, L.S., C.-W.L. and B.O.R.; validation, L.S., C.-W.L. and B.O.R.; investigation, L.S., C.-W.L. and B.O.R.; resources, A.G.; writing—original draft preparation, L.S., C.-W.L. and B.O.R.; writing—review and editing, A.G., L.S., C.-W.L. and B.O.R.; visualization, L.S., C.-W.L. and B.O.R.; supervision, A.G.; project administration, A.G.; funding acquisition, A.G. All authors have read and agreed to the published version of the manuscript.

**Funding:** This work was supported by the Office of Naval Research under Grant No. N00014-18-1-2536, the National Science Foundation under the Grant Opportunities for Academic Liaison with Industry (GOALI) program under Grant 1807940, the National Science Foundation Graduate Research Fellowship Program under Grant No. DGE 1256260, and the Chia-Lun Lo Fellowship at the University of Michigan.

**Institutional Review Board Statement:** Not applicable.

**Informed Consent Statement:** Not applicable.

**Data Availability Statement:** Not applicable.

**Conflicts of Interest:** The authors declare no conflict of interest.

## Appendix A

Here, the approximations that are used to calculate the quality factor of a metasurface with electrically thin spacers are derived. First, we will show that the internal matching resistance, $R_{int}$, is independent of the spacer's dielectric constant, $\varepsilon_r$. For an electrically thin spacer $\beta d << 1$. Using second-order small angle approximations, $\sin \beta d \approx \beta d$ and $\cos \beta d \approx 1 - \frac{(\beta d)^2}{2}$, in the expression for $R_{int}$ yields,

$$R_{int} = \frac{Z_{in} + Z_L + \sqrt{Z_{in}Z_L}\cos\phi_{21}}{\sin^2\phi_{21}} \frac{(\omega\mu_0 d)^2}{Z_{in}Z_L}. \tag{A1}$$

Which, does not depend on the dielectric constant of the spacer so $R_{int}$ is independent of the spacer's dielectric constant.

Next, we will derive the approximate expressions for the impedance sheets (31)–(33). Converting the $\sin(2\beta d)$ term in (32) to $2\sin(\beta d)\cos(\beta d)$ and making the same small argument approximations in (31)–(33),

$$Z_{s1} = -j\frac{\omega\mu_0 d}{1 - \frac{\omega^2\mu_0\varepsilon_r\varepsilon_0 d^2}{2} + (\frac{X_{12}+X_{22}}{\det Z})\omega\mu_0 d} \tag{A2}$$

$$Z_{s2} = -j\frac{(\omega\mu_0 d)^2 X_{12}}{\det Z + 2X_{12}\omega\mu_0 d(1 - \frac{\omega^2\mu_0\varepsilon_r\varepsilon_0 d^2}{2})} \tag{A3}$$

$$Z_{s3} = -j\frac{\omega\mu_0 d}{1 - \frac{\omega^2\mu_0\varepsilon_r\varepsilon_0 d^2}{2} + (\frac{X_{12}+X_{11}}{\det Z})\omega\mu_0 d}. \tag{A4}$$

Because we only need to consider capacitive sheets in the calculation for the quality factor, we can set (A2)–(A4) equal to the impedance of a capacitive element $Z_{si} = 1/j\omega C_{si}$. The following expressions are produced by setting (A2)–(A4) equal to $1/j\omega C_{si}$ and solving for the sheet capacitances,

$$C_{s1} = \frac{1}{\omega}\left(\frac{1}{\omega\mu_0 d} + \frac{X_{22}+X_{12}}{\det(Z)}\right) - \frac{\varepsilon_r\varepsilon_0 d}{2} \tag{A5}$$

$$C_{s2} = \frac{1}{\omega}\left(\frac{2}{\omega\mu_0 d} + \frac{\det(Z)}{X_{12}}\frac{1}{(\omega\mu_0 d)^2}\right) - \varepsilon_r\varepsilon_0 d \tag{A6}$$

$$C_{s3} = \frac{1}{\omega}\left(\frac{1}{\omega\mu_0 d} + \frac{X_{11}+X_{12}}{\det(Z)}\right) - \frac{\varepsilon_r\varepsilon_0 d}{2}. \tag{A7}$$

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
