# Peer review of "Fundamentals of Lossless, Reciprocal Bianisotropic Metasurface Design"

_photonics, doi:10.3390/photonics8060197_

Round 1

Reviewer 1 Report

The paper entitled "Fundamentals of lossless, reciprocal bianisotropic metasurface design" reports the fundamental theory and systematic design principles of lossless and reciprocal bi-isotropic and bi-anisotropic metasurface. The paper is well written and presented and is found suitable for publication. Some suggestions are given below:

  1. I would suggest the authors to include some analysis taking the permittivity of the multilayer structure into account. Meaning how the relative permittivity of the substrate affects the bandwidth of the metasurface. Is there any trade-off between the permittivity values used in the design and the achievable bandwidth?  

2. In the design examples, the metasurface is realized using dispersive impedance sheet/anisotropic boundary conditions in HFSS. How much will the performance deviate when the metasurface is practically realized using the canonical elements such as metallic dipoles, as it is sometimes hard to achieve the required impedance/admittance values. A brief statement on this will suffice.

Reviewer 2 Report

This paper introduced the notion of a metasurface quality factor for three-sheet metasurfaces, which can be used to predict the bandwidth of a homogeneous metasurface and to locate problematic unit cells with several design examples. This manuscript is interesting and novel, and I can recommend it for publication in Photonics.

Author Response

No questions to answer.

Reviewer 3 Report

The authors realize several examples of practical devices with phase and polarization control  based on bi-anisotropic metasurface designs. I found the paper very interesting. The manuscript is well written and organized nicely. The obtained results seem sound to me. Based on these points, I think the paper deserve publication in this journal. 

Author Response

No questions to answer.